**Data Availability Statement:** All relevant data are within the manuscript.

# Randomized experimental population-based study to evaluate the acceptance and completion of and preferences for cervical cancer screening

Marcela Vieira Lordelo[1], Cleyton Zanardo Oliveira[2], Luisa Aguirre Buexm[1], Rui Manuel Vieira Reis[1,3], Adhemar Longatto-Filho[1,3,4,5], Júlio César Possati-Resende[6], Fabiana de Lima Vazquez[1,6]*, José Humberto Tavares Guerreiro Fregnani[1,7]

1 Molecular Oncology Research Centre, Barretos Cancer Hospital, Barretos, Brazil, 2 Biostatistics, Centre for Teaching and Research, Beneficência Portuguesa de São Paulo, São Paulo (SP), Brazil, 3 Institute of Life and Health Sciences (ICVS), University of Minho, Braga, Portugal, 4 Faculty of Medicine, Department of Pathology, Medical Laboratory of Medical Investigation (LIM) 14, University of São Paulo, São Paulo, Brazil, 5 ICVS/3B's, Associated Laboratory of the Government of Portugal, Braga/Guimarães, Portugal, 6 Prevention Department, Barretos Cancer Hospital, Barretos, Brazil, 7 Education, A.C. Camargo Cancer Centre, São Paulo, Brazil

* fabilivazquez@gmail.com

## Abstract

Cervical cancer has high incidence and mortality rates, especially in less-developed countries. Prevention methods are well established, but there are still barriers preventing some Brazilian women from undergoing a Pap sample. The objective of the study was to evaluate the acceptance, preferences and completion of four screening methods. This has an experimental design (community trial). A total of 164 participants who had never had a Pap sample or had not had one for more than three years were included. The city's urban area was stratified by census tracts and divided according to income and education levels. Women belonging to the lower-income strata were considered in the study. Random blocks were numbered into five intervention groups (Group 1- Pap sample at the hospital; Group 2- Pap sample in the mobile unit; Group 3- urine self-collection; Group 4- vaginal self-collection; Group 5- woman's choice). Only 164 women met all of the eligibility criteria (15.3%). Most of them accepted the assigned method (92%), but only 84% of the women completed the collection step. The acceptance rates were as follows: Group 1 (100%), Group 2 (64.5%), Group 3 (100%) and Group 4 (91.4%). In Group 5, the women's preferences were distributed as follows: examination performed at the hospital, 13 women (33.3%); examination performed at the mobile unit, 11 women (28.2%); urine self-collection, 11 women (28.2%); and vaginal self-collection, 4 women (10.3%). This study suggests that methods that allow cervical sampling collected near the women's domicile might improve the acceptance and completion of preventive tests. This finding is relevant for the development of new cervical cancer screening strategies.

**Funding:** The author(s) received no specific funding for this work.

**Competing interests:** NO authors have competing interests.

## Introduction

Cervical cancer ranks seventh in incidence (3.1%) and mortality (3.3%) among cancers worldwide [1]; this ranking includes all cancers except for nonmelanoma skin cancer. It is the second leading cause of death in 36 of 185 studied countries, behind only breast cancer [2]. Approximately 604,000 new cases and 342,000 deaths due to the disease are recorded annually [1]. In Brazil, it is the third leading cause of cancer death in women [3].

Despite the efforts of the Health Public Authorities to improve the coverage rates of Pap samples, the incidence and mortality rates from cervical cancer remain high, often due to late diagnosis. Barriers still exist due to either resistance to undergoing the exam or structural difficulties, such as a lack of supplies and training for professionals. The acceptance of Pap samples is extremely important to achieve positive results. Some limitations to performing a Pap sample have been previously reported, such as discomfort, shame, fear of pain, lack of time and lack of a reliable professional to perform the exam [4,5]. Molecular testing is a better-accepted alternative to screening programmes [6,7].

The use of the human papillomavirus (HPV) test in combination with cytology testing in screening programmes has been described and is recommended for women over 30 years of age [8]. It can complement other current methods, such as self-collection of vaginal samples, and can increase coverage in places that do not have easy access to Pap samples, since women in these areas will be screened only a few times in their lives and the high sensitivity of HPV testing is important and may increase the acceptability and effectiveness of screening programmes [7,9–12]. The results of these negative molecular tests are more reliable than negative cytology results and can assist in the monitoring of women with screening programmes. These HPV tests combined with the results of a Pap sample, if performed, may lead to an 8-year reduction in mortality from this cancer [7]. They can be adopted as a primary screening strategy [13] alone or in combination with Pap samples.

To the best of our knowledge, few studies have compared different less-invasive sampling methods in the Brazilian population and analysed reasons for some particular preferences among the methods [13]. Acceptance may increase when women have choices. Such information could supposedly help mitigate the structural and organizational barriers and weaknesses of the Cervical Cancer Prevention Programme and reinforce adherence. The objective of this study was to evaluate the acceptance and completion of and preference for four screening methods.

## Materials and methods

### Study design and sample size

This was an experimental open-label study with a community trial conducted in the city of Barretos, state of São Paulo, Brazil. It took place from November 2018 to January 2020. Barretos is divided into census tracts, which are units of territory designed for registration control that have blocks of houses and/or land with dimensions and numbers that can be registered by a qualified professional, as defined by the Brazilian Institute of Geography and Statistics (IBGE) [14]. Barretos contains 148 census tracts. A total of 132 eligible census sectors were considered because they included the necessary information, such as the economic activity in the region (agriculture, industry, trade and services), to obtain the income of the heads of household.

The census tracts were divided into nine strata coded with letters A through I according to the income and education levels of the heads of household, using income as reference. Women living in the lower-income strata for heads of household (A, B and C) were included. Those strata include families that earn up to half a minimum wage per person or up to 3 minimum wages as their total monthly income, considering the minimum wage of R $998.00 effective on

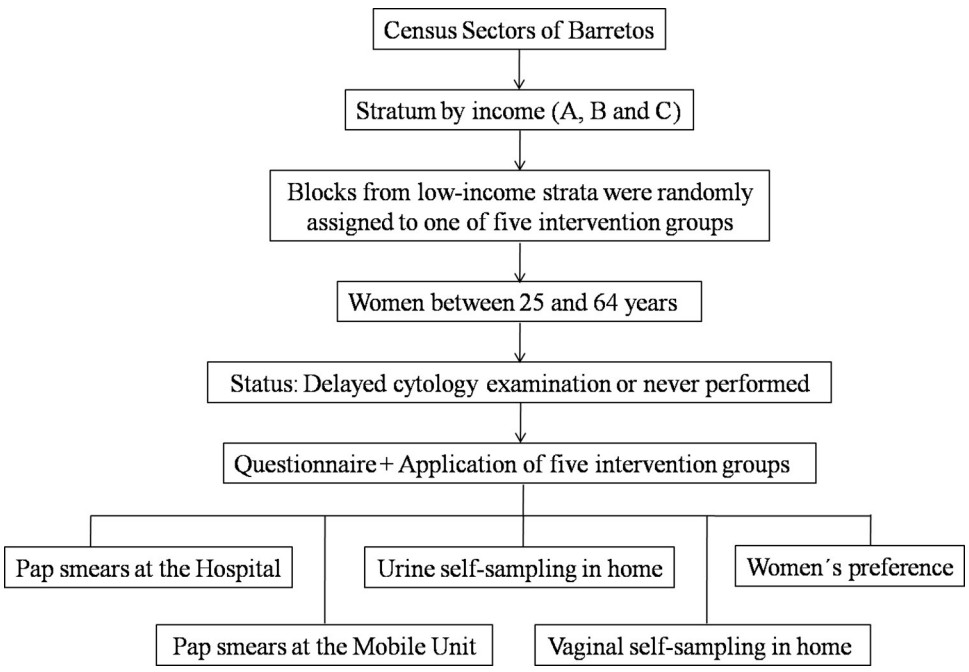

**Fig 1. Summarizes the study methodology developed by the researchers.** Each block was allocated to an intervention group: Group 1- performance of a Pap sample at the Cancer Prevention Department of Barretos Cancer Hospital (HCB), Brazil; Group 2- performance of a Pap sample in the mobile unit by the researcher; Group 3- urine self-collection by the participant at the time of inclusion; Group 4- self-collection of vaginal material by the participant at the time of inclusion; Group 5- the participant selected her preferred method from among the four options. Randomly selected households that had no eligible participants, that refused to participate or had no one living in the household were excluded from the study.

January 1, 2019, according to Decree 9,661 [15,16]. The low-income population was chosen because of its increased likelihood of resistance to undergoing a Pap sample, lack of knowledge and difficulty in accessing health services. In each stratum, the blocks (several houses) were numbered and randomly assigned to intervention groups by simple random computer-based sampling to ensure the appropriate number of women in each intervention group. R software was used. The researcher performed the inclusions on the block knowing which intervention group he belonged to, to offer the participant at the end of the interview. For each household, only one woman could participate in the study (Fig 1).

## Inclusion and exclusion criteria

The criteria established for the inclusion of participants in the study were as follows: sexually active women; never having undergone a Pap sample or not having done so for more than three years; age between 25 and 64 years; living in low-income areas; and able to be interviewed at domicile. The criteria established for the exclusion of patients from the study were as follows: previous total hysterectomy; current pregnancy; residing in Barretos for less than a year; and difficulty understanding the questionnaire.

## Data and sample collection

The questionnaire addressed questions related to health and socioeconomic and demographic information. At the end of the interview, an intervention method was offered to the participant based on the group to which the block where they lived was assigned.

Households on the selected blocks were visited by the researcher with a mobile unit driven by the hospital's driver. All the materials used in the different intervention groups were stored in the mobile unit's cabinets.

In Group 1, the participants were invited to undergo a Pap sample and HPV screening at HCB. On the scheduled day, the researcher checked with Enterprise Resource Planning to determine whether the study participant had been seen at the hospital and identify her as a study participant; when necessary, the women were referred for treatment at the institution as a routine follow-up. They received a letter inviting them to undergo a Pap sample at the hospital along with the appropriate instructions, as indicated by the Care Protocols of the Departments of Prevention and Oncological Gynaecology of the HCB based on guidelines provided by the Brazilian Ministry of Health [17].

In Group 2, the participants were seen by the researcher in the mobile unit (which had been adapted for performing gynaecological examinations) after the interview to undergo a Pap sample and HPV screening.

In Group 3, the participants were asked to self-collect urine at the domicile and received a kit and detailed verbal instructions on how to use it from the researcher. The urine sample was deposited in the sterile collector, collected after the interview, prepared inside the mobile unit by adding 10 millilitres of biological material preservative (EDTA) and stored in a thermal box until arrival at the HCB for HPV screening. The women received a letter inviting them to undergo a Pap sample at the hospital along with the appropriate instructions.

In Group 4, the participants were asked to self-collect a vaginal sample at the domicile and received a kit with detailed instructions for use immediately after the interview. The material was stored in a SurePath Medium container (BD, Burlington, USA) and sent to the hospital on the same day. The women received a letter inviting them to undergo a Pap sample at the hospital along with the appropriate instructions.

In Group 5, the participants had the opportunity to choose the method they preferred from among the four methods described above.

For all groups, HPV molecular screening was performed. Aliquots of the material in the flasks were analysed in SurePath™ using the Cobas HPV test (Roche Molecular Systems, USA), which is available at the hospital's Research Centre for Molecular Oncology. For the urine samples, a six-millilitre aliquot of urine was added to 10 ml of a 50 mM of EDTA solution to perform the HPV DNA test. The test protocol was performed as described by the manufacturer.

The Cobas platform is an automated amplification device (using real-time PCR-polymerase chain reaction) for the detection of 14 HPV genotypes with high oncogenic risk (16, 18, 31, 33, 35, 39, 45, 51, 52, 56, 58, 59, 66 and 68) and the genotyping of HPV16 and HPV18 types [18]. The DNA was extracted automatically by the Cobas x480 instrument, and then the samples were transferred to the Cobas z480 for DNA amplification. Interpretation, which was the last stage, was performed using Cobas 4800 software.

The Brazilian Ministry of Health still indicates the Pap sample as a method of choice for cervical cancer screening [17]. The samples were analysed at the Department of Pathology of the HCB. Scraped cervical tissue was transferred to a flask with preservative solution (SurePath™ Preservative Fluid, Becton & Dickinson, USA). Slides with cervical tests and staining were automatically produced using the PrepMate™/PrepStain™ system (Becton & Dickinson, USA), a procedure that was duly standardized at the hospital. The cytological findings were classified according to the Bethesda System criteria [19].

According to the study by Castle et al.[20], considering 60% completion by the group who underwent the Pap sample at the health unit and 100% completion by the vaginal self-collection group, an alpha error of 5% and a beta of 10%, the estimated sample size was 30

participants per group. The drop-out rate was 10%, with 33 participants per group. The platform used to measure the sampling power and significance was GPower 3.1.

## Ethical consideration

This study was approved by the local Research Ethics Committee and the Brazilian National Research Ethics Committee (CAAE 56755116.7.0000.5437). When the women were approached about participating, they were given a brief explanation of the project in simplified language and asked if they could answer some questions. The questions were presented using electronic instruments (tablets) loaded with REDCap software (Research Electronic Data Capture). All participants provided written informed consent, and all personal information was kept encrypted in a database to ensure data confidentiality.

## Statistical analysis

Descriptive analysis was used to characterize the sample. The data are described using contingency tables with absolute and relative frequencies. Fisher's exact test was used to assess the association between the variables of interest and intervention groups. The Statistical Package for Social Science for Windows—SPSS (version 21.0) was used for the statistical analysis of the data, and a significance level of 5% was considered in all tests. The data were collected and stored on the HCB's REDCap platform, following the general data protection regulations.

## Results

The field research was conducted in the three low-income strata; 3,313 households were visited, and 1,070 participants were contacted (i.e., approached during the field research). The study included a total of 164 women. Some of the women met more than one exclusion criterion. Fig 2 summarizes the composition of the sample according to the research stage and intervention group.

The mean age of the participants was 43.74 years (SD: 11.4; 25–64). Table 1 shows the sociodemographic characteristics by intervention group; there were no significant differences among the groups.

Table 2 shows the acceptance and completion rates by the intervention group. Acceptance was 100% in the group assigned to undergo a Pap sample at the hospital and the group that performed urine self-collection at the domicile. The acceptance rate was 64.5% in the group assigned to undergo a Pap sample at the mobile unit and 91.4% in the group assigned to perform vaginal material self-collection. There was a significant difference in rate among the groups, with Group 2 exhibiting lower acceptance than the other groups. The completion rate was 100% in the group assigned to perform urine self-collection at the domicile. The completion rate was 53.3% for the group assigned to undergo a Pap sample at the hospital, 90% for the group assigned to undergo a Pap sample at the mobile unit and 93.8% for the vaginal sample self-collection group. Two participants in the mobile unit group and two participants in the vaginal sample self-collection group were unable to complete the procedure due to physical limitations and reports of pain. In the analysis by the intervention group, none of the evaluated variables was associated with the outcome of interest. There was a significant difference in rates among the groups, with the group assigned to undergo a Pap sample at the hospital exhibiting a lower completion rate than the others ($p<0.001$, Fisher's exact test).

In Group 5, the participant preference group, 100% acceptance of one of the available methods was observed. Pap sample at the hospital was chosen by 13 participants (33.3%); 11 chose to undergo a Pap sample at the mobile unit (28.2%); 11 preferred to perform urine self-collection at domicile (28.2%), and 4 women opted for vaginal material self-collection at domicile

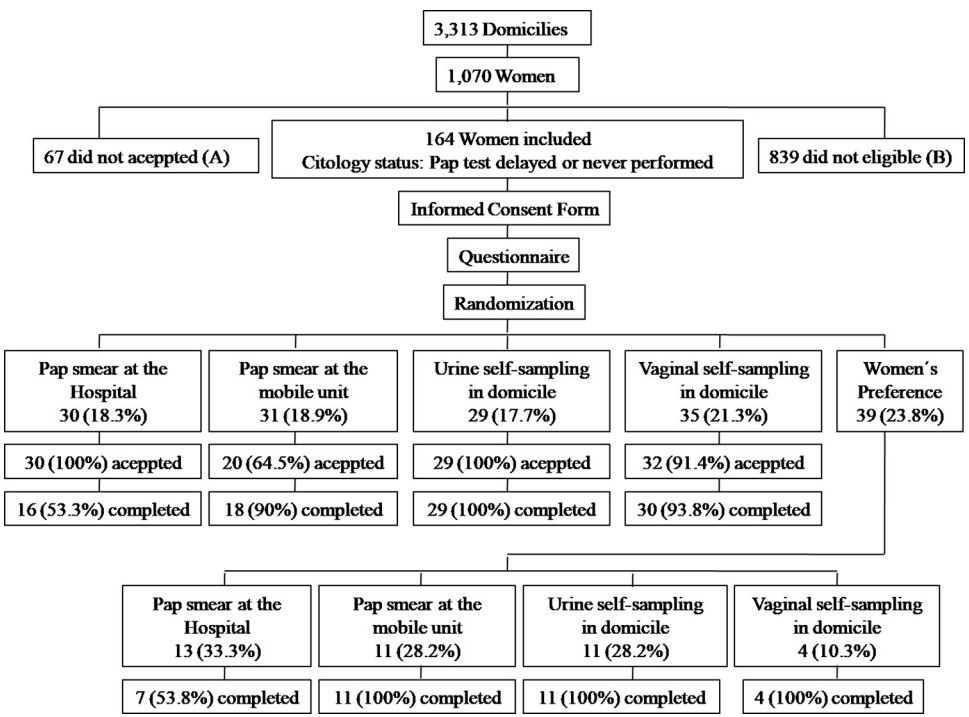

**Fig 2. Composition of the study population according to the stage of the field research.** (A) Refusal due to lack of time (n = 52); refusal because they did not want to talk about the subject (n = 8); refusal due to a current health problem (n = 5) and refusal for reasons not defined (n = 2). (B) The reasons for exclusion from the study were Pap sample performed less than three years ago (n = 400); age below 25 years (n = 65); residency in Barretos for less than a year (n = 17); pregnancy (n = 18); age over 64 years (n = 179); history of total hysterectomy (n = 156); and inability to answer the questions (n = 1).

(10.3%). Of the women who chose to schedule a Pap sample at the hospital, only 7 women (53.8%) attended the exam. For the other three methods, all women followed through to completion.

## Discussion

Cervical cancer remains a global health issue, especially in less developed countries, where preventive exams have low coverage [21–23]. This study sought to evaluate the acceptance of four cervical cancer screening methods, as well as to understand the complexity behind the entire completion and the reasons related to their preferred option to choose one of them. The highest acceptance rate was found in the group assigned to undergo Pap sample collected at the hospital ambulatory, followed by the urine self-collection group and the group that allowed the participant to choose her preferred method. The group assigned to have a Pap sample at the hospital showed the lowest completion rate; among the women who chose to go undergo this method, only 54% followed through to completion. Shame was the main barrier that women reported for not undergoing a routine Pap sample.

In the first group, all women who were asked to undergo a Pap sample at the hospital agreed to the screening by scheduling it for a day of their choosing. Therefore, a 100% acceptance rate was achieved. Regarding the completion rate, however, only 53.3% (16/30) went to the hospital to complete the test. These results differ from those observed in other countries such as the United States, Malaysia, Ghana and Norway, with reported completion rates of 40.5% (17/42), 41.8% (132/316), 60.4% (228/377) and 66% (863,042/1,309,679), respectively [24,25]. Brazilian

**Table 1. Sociodemographic data of the study participants by intervention group.**

| Variable | Scheduling at the hospital (n = 30) | Collection at the mobile unit (n = 31) | Urine self-collection (n = 29) | Vaginal material self-collection (n = 35) | Participant preference (n = 39) |
|---|---|---|---|---|---|
| **Age (years)** | | | | | |
| 25–40 | 15 (50.0%) | 14 (45.2%) | 11.(37.9%) | 15 (42.9%) | 19 (48.7%) |
| 41–64 | 15 (50.0%) | 17 (54.8%) | 18.(62.1%) | 20 (57.1%) | 20 (51.3%) |
| **Marital status** | | | | | |
| Single | 5 (16.7%) | 6 (19.4%) | 5.(17.2%) | 8 (22.9%) | 11 (28.2%) |
| Married/stable union | 22 (73.3%) | 21 (67.7%) | 19.(65.5%) | 23 (65.7%) | 24 (61.5%) |
| Divorced/widowed/ other | 3 (10.0%) | 4 (12.9%) | 5.(17.2%) | 4 (11.4%) | 4 (10.3%) |
| **Education level** | | | | | |
| Illiterate | 0 (0.0%) | 1 (3.2%) | 1.(3.4%) | 0 (0.0%) | 2 (5.1%) |
| Primary education | 20 (66.7%) | 13 (41.9%) | 11.(37.9%) | 17 (48.6%) | 14 (35.9%) |
| Secondary education | 5 (16.7%) | 11 (35.5%) | 13.(44.8%) | 14 (40.0%) | 14 (35.9%) |
| Higher education | 5 (16.7%) | 6 (19.4%) | 4.(13.8%) | 4 (11.4%) | 9 (23.1%) |
| **Income level** | | | | | |
| Up to 1 minimum wage | 5 (16.7%) | 6 (19.4%) | 6.(20.7%) | 5 (14.3%) | 11 (28.2%) |
| 1–3 minimum wages | 21 (70.0%) | 18 (58.1%) | 19.(65.5%) | 23 (65.7%) | 21 (53.8%) |
| >3 minimum wages | 3 (10.0%) | 6 (19.4%) | 4.(13.4%) | 7 (20.0%) | 5 (12.8%) |
| Did not respond | 1 (3.3%) | 1 (3.2%) | 0.(0.0%) | 0 (0.0%) | 2 (5.1%) |

data from the study by Castle et al.[20] showed a completion rate of 60%;the authors reported that after three invitations, this value increased to 75% completion. These formerly reported acceptance/completion rates vary from those of our study, with some studies reporting higher values and others reporting lower values. Social and local factors may influence women's ability to travel to obtain health services.

The participants who were invited to undergo the exam in the mobile unit had the opportunity to complete the examination right outside their door, with all the available resources; it was expected that the convenience of not having to travel to complete the exam would increase acceptance. However, the acceptance rate was merely 64.5% (20/31), and the completion rate was 90% (18/20). This result may be related to nongeographic barriers, such as resistance,

**Table 2. Acceptance and completion rates by intervention group.**

| | Pap sample at the hospital | Pap sample at the mobile unit | Urine self-collection in domicile | Vaginal self-collection in domicile | Woman's preference | Total | p-value [a] |
|---|---|---|---|---|---|---|---|
| **Accepted** | | | | | | | **<0.001** |
| No | 0 (0.0%) | 11 (35.5%) | 0 (0.0%) | 3 (8.6%) | 0 (0.0%) | 14 (8.5%) | |
| Yes | 30 (100%) | 20 (64.5%) | 29 (100%) | 32 (91.4%) | 39 (100%) | 150 (91.5%) | |
| Total | 30 (100%) | 31 (100%) | 29 (100%) | 35 (100%) | 39 (100%) | 164 (100%) | |
| **Completed** | | | | | | | **<0.001** |
| No | 14 (46.7%) | 2 (10%) | 0 (0.0%) | 2 (6.3%) | 6 (15.4%) | 24 (16%) | |
| Yes | 16 (53.3%) | 18 (90%) | 29 (100%) | 30 (93.8%) | 33 (84.6%) | 126 (84%) | |
| Total | 30 (100%) | 20 (100%) | 29 (100%) | 32 (100%) | 39 (100%) | 150 (100%) | |

[a]Fisher's exact test.

shame and lack of knowledge of the subject among women [26,27]. In the study by Awua et al. [28],the acceptance rate was 95.1% for exams performed near the participant's domicile. The authors used a strategy that was similar to that used in this study but differed by providing the opportunity for the woman to choose the day she would undergo the test; it is believed that was the reason for the greater acceptance rate.

All participants who were offered the opportunity to use the urine self-collection method accepted and completed it. It is important to highlight some points that probably influenced this result, namely, ease of execution and familiarity with the test. The results of other studies using this method converge with the data of the present study as the importance of acceptance by women is clear. Sabeena et al [29]. conducted domicile visits in rural areas of India offering this method and reported a participant acceptance rate of 98.7%. Sy et al [30] conducted a community research project that found an acceptance rate of 95% for urine self-collection among women. Geographic and cultural differences may influence women's acceptance of various methods.

In the vaginal material self-collection group, 91.4% (32/35) of the participants agreed to perform the method, and 93.8% (30/32) completed it. Resende et al [31], who conducted a study at the same institution with residents in the rural community, offered a vaginal material self-collection method to participants and obtained 95.6% acceptance. Some studies have shown better acceptance for the collection of samples at domicile and have reported that this method reaches women resistant to Pap samples and places with little resources [10,32–34]. When this method was offered by community health agents during domicile visits, a fourfold increase in the acceptance of screening was found [35]. In Brazil, for example, the training of community health professionals could significantly improve coverage [20,36].These data reinforce the efficacy of this method in attempts to reduce the incidence and mortality of cervical cancer [37].

In the participant's preference intervention group, a greater number of women chose to undergo a Pap sample at the hospital, but only 53.8% (7/13) completed the process. These completion rates suggest that choosing methods that are performed close to domicile facilitates the completion of the test. Methods that can be offered at a domicile visit encourage greater acceptance and completion [28,32,35]. Some barriers include a lack of knowledge of how to perform vaginal material self-collection by the Brazilian population. It is believed that lack of knowledge is the reason that the method was not chosen as often as the others, whose terms and methods of execution are better known.

It is worth noting that evidence currently supports the possibility of using specific biomarkers to identify cervical cancer early, providing a better prognosis for women [38,39]. Emerging data suggests that high-grade cervical dysplasia in HPV-negative patients has better outcomes than in patients with documented high-risk HPV infection [40].

The results herein reported must be considered with caution due to the limitations imposed by the study design. Firstly, the casuistry size is not sufficient for some statistical analyses. Secondly, in the participant's preference group, the women who had previously undergone a Pap sample may have had different preferences from those who never had received one. Thirdly, the lack of knowledge about how to self-sampling vaginal material might be collected introduced limited acceptability for some women to opt for this method. Fourthly, the middle- and high-income strata were not included, but this could be a strength since the groups included, i.e., low-income and unscreened women, are a minority in the local population and have a minor degree of medical procedures information.

## Conclusion

This study observed that all participants agreed to schedule a Pap sample in the hospital, but there was a low percentage of completion. For the urine self-collection method, all procedures

were accepted and performed. For the vaginal material self-collection group, there was good acceptance and completion. The lowest acceptance was found for the mobile unit intervention group, but the completion rate was good. All women who chose a method performed near their domicile (mobile unit) or at their domicile (urine self-collection and vaginal self-collection) completed the collection of material. These data call attention to the importance of adopting personalized measures to ensure that the testis performed correctly and during the most acceptable period, especially for women who are resistant to cervical cancer screening and introduce educational methods for cancer prevention for the population with low resources of health information.

## Acknowledgments

The authors thank Philip E. Castle for the rich discussions on prevention, which motivated the design of this study. The authors thank the following organizations: Department of Cancer Prevention, Department of Molecular Oncology, Research Support Centre and Department of Pathology, Barretos Cancer Hospital, and the Public Ministry of Labor Campinas (Research, Prevention, and Education of Occupational Cancer) They also thank Camilla Martins, Elisa Messias, Karen Borba, Larissa Silva, Marcos Lima and Viviane Andrade from the Research Support Centre.

## Author Contributions

**Conceptualization:** Marcela Vieira Lordelo, Cleyton Zanardo Oliveira, Rui Manuel Vieira Reis, Adhemar Longatto-Filho, Júlio César Possati-Resende, Fabiana de Lima Vazquez, José Humberto Tavares Guerreiro Fregnani.

**Data curation:** Marcela Vieira Lordelo, Fabiana de Lima Vazquez, José Humberto Tavares Guerreiro Fregnani.

**Formal analysis:** Marcela Vieira Lordelo, Fabiana de Lima Vazquez, José Humberto Tavares Guerreiro Fregnani.

**Funding acquisition:** Rui Manuel Vieira Reis.

**Investigation:** Marcela Vieira Lordelo, José Humberto Tavares Guerreiro Fregnani.

**Methodology:** Cleyton Zanardo Oliveira, Júlio César Possati-Resende, Fabiana de Lima Vazquez, José Humberto Tavares Guerreiro Fregnani.

**Project administration:** Fabiana de Lima Vazquez, José Humberto Tavares Guerreiro Fregnani.

**Supervision:** Fabiana de Lima Vazquez, José Humberto Tavares Guerreiro Fregnani.

**Writing – original draft:** Marcela Vieira Lordelo, Fabiana de Lima Vazquez, José Humberto Tavares Guerreiro Fregnani.

**Writing – review & editing:** Marcela Vieira Lordelo, Cleyton Zanardo Oliveira, Luisa Aguirre Buexm, Rui Manuel Vieira Reis, Adhemar Longatto-Filho, Júlio César Possati-Resende, Fabiana de Lima Vazquez, José Humberto Tavares Guerreiro Fregnani.

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
