## [Decision Letter · Decision Letter 0]

3 Jan 2022

PONE-D-21-23169Randomized experimental population-based study to evaluate the acceptance and completion of and preferences for cervical cancer screeningPLOS ONE

Dear Dr. LORDELO,

Thank you for submitting your manuscript to PLOS ONE. After careful consideration, we feel that it has merit but does not fully meet PLOS ONE’s publication criteria as it currently stands. Therefore, we invite you to submit a revised version of the manuscript that addresses the points raised during the review process.

Editor's comments:

1. Please follow PLoS One's guidelines for the manuscript preparation (including the references).

2. Please cite (at least) five relevant articles from PLoS One.

3. The manuscript requires linguistic copy editing (please also submit the English copy-editing certificate).

4. Please revise the manuscript per the following reviewers' comments.

We look forward to receiving your revised manuscript.

Kind regards,

Farzad Taghizadeh-Hesary

Academic Editor

PLOS ONE

Journal Requirements:

Additional Editor Comments: 

Reviewer #1 comments:

I read with great interest the manuscript, which falls within the aim of this Journal. In my honest opinion, the topic is interesting enough to attract the readers’ attention. Nevertheless, authors should clarify some points and improve the discussion, as suggested below. Authors should consider the following recommendations: - Manuscript should be further revised in order to correct some typos and improve style. - To date, several lines of evidence support the possibility to use specific biomarkers to identify early stage cervical cancer and, in this way, offer a better prognosis to the patients. This point deserves to be discussed, referring to: PMID: 28918603; PMID: 28466456. - I would recommend to stress novel pieces of evidence about high-risk HPV-negative high-grade cervical dysplasia, which seems to have more favorable outcomes than patients with documented high-risk-HPV infection (PMID: 33514481) as well the potential role of pre-operative conization before radical hysterectomy for early-stage cervical cancer (PMID: 32320800).

Reviewer #2 comments:

The manuscript is well written but will need a few modifications. There are a few typographical and technical errors that have been highlighted in the attached documents. Please do well to create sub-headings for the conclusion and others as stated in the attached document (please find the attached word document). Good job

Reviewer #3 comments:

1) Abstract-line 4: please add the "completion" to the study objectives. 2) Line 80. Was "simple-random sampling" computer-based? 3) Please improve the resolution of figures. 4) 201-202: "The highest acceptance rate was found in the group assigned to undergo Pap sample collected at the hospital ambulatory, followed by the urine self collection group and the group that allowed the participant to choose her preferred method". But all three groups had same acceptance rate of 100%. Please correct the sentence. 5) The manuscript requires English copy editing. 6) In methods section, please mention that it was an open-label study.

Reviewers' comments:

Reviewer's Responses to Questions

**Comments to the Author**

1. Is the manuscript technically sound, and do the data support the conclusions?

Reviewer #1: Yes

Reviewer #2: Yes

Reviewer #3: Yes

2. Has the statistical analysis been performed appropriately and rigorously? 

Reviewer #1: Yes

Reviewer #2: No

Reviewer #3: I Don't Know

3. Have the authors made all data underlying the findings in their manuscript fully available?

Reviewer #1: Yes

Reviewer #2: Yes

Reviewer #3: Yes

4. Is the manuscript presented in an intelligible fashion and written in standard English?

Reviewer #1: Yes

Reviewer #2: Yes

Reviewer #3: No

5. Review Comments to the Author

Reviewer #1: I read with great interest the manuscript, which falls within the aim of this Journal. In my honest opinion, the topic is interesting enough to attract the readers’ attention. Nevertheless, authors should clarify some points and improve the discussion, as suggested below.

Authors should consider the following recommendations:

- Manuscript should be further revised in order to correct some typos and improve style.

- To date, several lines of evidence support the possibility to use specific biomarkers to identify early stage cervical cancer and, in this way, offer a better prognosis to the patients. This point deserves to be discussed, referring to: PMID: 28918603; PMID: 28466456.

- I would recommend to stress novel pieces of evidence about high-risk HPV-negative high-grade cervical dysplasia, which seems to have more favorable outcomes than patients with documented high-risk-HPV infection (PMID: 33514481) as well the potential role of pre-operative conization before radical hysterectomy for early-stage cervical cancer (PMID: 32320800).

Reviewer #2: The manuscript is well written but will need a few modifications.

There are a few typographical and technical errors that have been highlighted in the attached documents.

Please do well to create sub-headings for the conclusion and others as stated in the attached document.

Good job

Reviewer #3: 1) Abstract-line 4: please add the "completion" to the study objectives.

2) Line 80. Was "simple-random sampling" computer-based?

3) Please improve the resolution of figures.

4) 201-202: "The highest acceptance rate was found in the group assigned to undergo Pap sample collected at the hospital ambulatory, followed by the urine self collection group and the group that allowed the participant to choose her preferred method". But all three groups had same acceptance rate of 100%. Please correct the sentence.

5) The manuscript requires English copy editing.

6. PLOS authors have the option to publish the peer review history of their article (what does this mean?). If published, this will include your full peer review and any attached files.

Reviewer #1: No

Reviewer #2: No

Reviewer #3: No

---

## [Author Response · Author response to Decision Letter 0]

24 Oct 2023

Excellent Sir

Farzad Taghizadeh-Hesary

Academic Editor

PLoS One

Dear Sir,

We hereby forward the corrected and revised manuscript entitled "Randomized experimental population-based study to evaluate the acceptance and completion of and preferences for cervical cancer screening", authored by Marcela Vieira Lordelo, Cleyton Zanardo Oliveira, Luisa Aguirre Buexm, Rui Manuel Vieira Reis, Adhemar Longatto-Filho, Júlio César Possati-Resende, Fabiana Lima Vazquez, José Humberto Tavares Guerreiro Fregnani, for your consideration.

First of all, we would like to thank the reviewers of this study who provided valuable suggestions and very pertinent comments, thus contributing to improving the quality of our article. After a detailed analysis of the comments and questions, as well as the errors pointed out and suggestions contained in the opinions sent to us, the article has undergone some changes, which are indicated below.

Thank you in advance for your attention,

Fabiana de Lima Vazquez

Researcher, Molecular OncologyResearch Centre, Barretos Cancer Hospital, Brazil

---

## [Editor Report · Decision Letter 1]

12 Jun 2024

Randomized experimental population-based study to evaluate the acceptance and completion of and preferences for cervical cancer screening

PONE-D-21-23169R1

Dear Dr. Vazquez,

We’re pleased to inform you that your manuscript has been judged scientifically suitable for publication and will be formally accepted for publication once it meets all outstanding technical requirements.

Kind regards,

Lucy W. Kivuti-Bitok, Ph.D.

Academic Editor

PLOS ONE
---

## [Editor Report · Acceptance letter]

2 Aug 2024

PONE-D-21-23169R1 

PLOS ONE

Dear Dr. Vazquez, 

I'm pleased to inform you that your manuscript has been deemed suitable for publication in PLOS ONE. Congratulations! Your manuscript is now being handed over to our production team.

Kind regards, 

on behalf of

Prof Lucy W. Kivuti-Bitok 

Academic Editor

PLOS ONE